# Shape selection and mis-assembly in viral capsid formation by elastic frustration

Carlos I Mendoza[1]*, David Reguera[2,3]

[1]Instituto de Investigaciones en Materiales, Universidad Nacional Autónoma de México, México, Mexico; [2]Departament de Física de la Matèria Condensada, Universitat de Barcelona, Barcelona, Spain; [3]Universitat de Barcelona Institute of Complex Systems (UBICS), Universitat de Barcelona, Barcelona, Spain

**Abstract** The successful assembly of a closed protein shell (or capsid) is a key step in the replication of viruses and in the production of artificial viral cages for bio/nanotechnological applications. During self-assembly, the favorable binding energy competes with the energetic cost of the growing edge and the elastic stresses generated due to the curvature of the capsid. As a result, incomplete structures such as open caps, cylindrical or ribbon-shaped shells may emerge, preventing the successful replication of viruses. Using elasticity theory and coarse-grained simulations, we analyze the conditions required for these processes to occur and their significance for empty virus self-assembly. We find that the outcome of the assembly can be recast into a universal phase diagram showing that viruses with high mechanical resistance cannot be self-assembled directly as spherical structures. The results of our study justify the need of a maturation step and suggest promising routes to hinder viral infections by inducing mis-assembly.

*For correspondence:
cmendoza@iim.unam.mx

**Competing interests:** The authors declare that no competing interests exist.

## Introduction

Viruses are fascinating biological and nanoscale systems (*Douglas and Young, 2006*; *Wen and Steinmetz, 2016*). In the simplest cases, these tiny pathogens are formed by a chain of RNA or DNA encased in a protein shell, also known as *capsid*, made from multiple copies of a single protein (*Flint et al., 2004*). Despite this apparent simplicity, viruses are able to perform many complex functions which are essential in their replication cycle. One of the most amazing one is their ability to self-assemble with an unparalleled efficiency and precision.

In vivo, the capsid of most viruses assembles from its basic building blocks, which could be individual capsid proteins, dimers, trimers or capsomers (i.e. clusters of five or six proteins, which constitute the structural and morphological units of the shell). The resulting structure has a precise architecture, which in most cases is spherical with icosahedral symmetry (*Roos et al., 2010*). Several viruses assemble their capsid before packaging the genetic material. In addition, the proteins of many viruses have the ability to self-assemble in vitro, even in the absence of genetic material, forming empty capsids.

The mechanisms of viral assembly have been the subject of recent and interesting investigations (*Hagan and Chandler, 2006*; *Elrad and Hagan, 2008*; *Nguyen et al., 2009*; *Johnston et al., 2010*; *Perlmutter and Hagan, 2015*; *Hagan and Zandi, 2016*). The assembly of a curved empty shell with a well-defined geometry and precise arrangement of the building blocks is a non-trivial process that resembles 2D crystallization on a curved space (*Meng et al., 2014*; *Gómez et al., 2015*). It also shares similarities with the formation of other related structures such as colloidosomes (*Meng et al., 2014*; *Dinsmore et al., 2002*; *Manoharan et al., 2003*), carboxysomes (*Perlmutter et al., 2016*; *Garcia-Alles et al., 2017*) or clathrin-coated pits (*Mashl and Bruinsma, 1998*; *Kohyama et al., 2003*; *Giani et al., 2016*). Capsid formation occurs via a nucleation process driven by the favorable binding energy between capsid proteins (*Zandi et al., 2006*). At the right assembly conditions,

thermal fluctuations induce the formation of small partial shells that tend to redissolve unless they reach a minimum critical size. Beyond this size, the shell grows by the progressive binding of subunits. As growth continues, the energy penalty of the naturally curved structure, due to the inescapable presence of the rim and the accumulation of elastic energy, can be larger than the favorable binding energy. This generates a natural self-limiting mechanism for the formation of partial shells of a finite size that do not grow until closing (*Grason, 2016*). In fact, there are in vitro experimental evidences of apparently stable partial capsids (*Law-Hine et al., 2016*), that seems to contradict the instability of intermediates that follows from Classical Nucleation Theory (CNT) (*Zandi et al., 2006*).

Recently, there has been a lot of interest in geometric frustration and crystal growth on spherical templates (*Zandi et al., 2004*; *Luque et al., 2012*; *Meng et al., 2014*; *Gómez et al., 2015*; *Grason, 2016*; *Azadi and Grason, 2016*; *Paquay et al., 2017*; *Li et al., 2018*; *Panahandeh et al., 2018*). Most previous works have focused either on templated growth on the surface of a sphere (*Zandi et al., 2004*; *Luque et al., 2012*; *Meng et al., 2014*; *Gómez et al., 2015*; *Grason, 2016*; *Li et al., 2018*; *Panahandeh et al., 2018*) or on analyzing the optimal shape of the resulting shell from pure elastic considerations (*Paquay et al., 2017*; *Lidmar et al., 2003*; *Schneider and Gompper, 2007*; *Morozov and Bruinsma, 2010*; *Castelnovo, 2017*), ignoring the importance of the delicate interplay of other ingredients such as the line tension, the chemical potential or the preferred curvature on their global stability and their process of formation.

Here, we analyze the conditions and mechanisms leading to mis-assembly of empty viral capsids by elastic frustration, taking into account *all* these ingredients. We find that the outcome of the assembly depends on three scaled parameters that can be properly tuned to trigger the formation of non-spherical and open shells. Theoretical predictions obtained with the use of Classical Nucleation Theory including elastic contributions are confirmed qualitatively using Brownian Dynamics simulations of a simple coarse-grained model. The results of this work help to better understand viral assembly and might have important implications in: envisaging novel routes to stop viral infections by interfering with their proper assembly; determining the optimal conditions for the assembly of protein cages with the desired geometry and properties for nanotechnological applications (*Douglas and Young, 2006*); and justifying the potential presence of seemingly stable intermediates that have been observed in recent experiments (*Law-Hine et al., 2016*).

## Results

### Self-assembly of a curved elastic shell

The continuous description of the assembly of empty spherical viral capsids is based on Classical Nucleation Theory (CNT) (*Zandi et al., 2006*). In its standard version, the free energy of formation of a partial shell of area $S$ is seen as the competition of an energy gain driving the assembly, and a rim energy penalty, due to the missing contacts at the edge of the shell. Due to the curvature of the shell and the existence of a preferred angle of interaction between capsid proteins there is another ingredient that has to be considered in the energetics of capsid formation: the elastic energy. Accordingly, the free energy of formation of a partial capsid of area $S$ can be modeled as

$$\Delta G = -S\frac{\Delta\mu}{a_1} + \Lambda l(S) + G_e. \tag{1}$$

The first term represents the gain in free energy associated with the chemical potential difference $\Delta\mu$ between subunits in solution and in the capsid, being $a_1$ the area per subunit. (With this definition, a positive $\Delta\mu$ is required to promote assembly). The second term is the total line energy of the rim, given by the product of the line tension $\Lambda$ times its length $l(S)$. Finally, the third term $G_e = G_s + G_b$ is the elastic energy associated with the in-plane stress, $G_s$, and the bending, $G_b$, energies introduced by the curvature of the shell. Both elastic terms will be modeled using continuum elasticity theory. For the bending energy we will use the generalization of Helfrich's model for systems with non-zero spontaneous curvature introduced recently by *Castelnovo (2017)* (see the Appendix). For the in-plane elastic energy, we will use results from continuum elasticity theory for small deformations of thin plates, building up on recent work on the formation and growth of crystal domains of different shapes on curved surfaces (*Lidmar et al., 2003*; *Meng et al., 2014*; *Seung and Nelson, 1988*; *Morozov and Bruinsma, 2010*; *Grason, 2016*; *Paquay et al., 2017*; *Köhler et al.,*

*2016*; *Schneider and Gompper, 2007*; *Majidi and Fearing, 2008*; *Castelnovo, 2017*). Both stretching and bending terms depend on the particular structure of the growing shell. Four different cases will be analyzed: hexagonally-ordered spherical cap without defects; spherical cap with one or many defects; ribbon and cylinder (see *Figure 1*). The reason to consider these particular structures is that they represent the most advantageous shapes to release the unfavorable elastic energy. In addition, cylindrical shells also appear frequently as outcome of in vitro assembly experiments. However, it is important to stress that the considered structures do not form a complete set of deformations.

The relative importance of stretching versus bending contributions is controlled by a single dimensionless parameter: the Föppl-von Kármán number (FvK) defined here as $\gamma \equiv Y R_0^2 / \kappa$, where $Y$ is the two-dimensional Young's modulus, $R_0$ is the spontaneous radius of curvature and $\kappa$ is the bending modulus. Most previous studies have focused on the elastic energy and growth of crystals on top of a spherical template of fixed radius $R$. This case resembles the bending-dominated regime discussed below.

## Bending-dominated regime

In the limit $\gamma = Y R_0^2 / \kappa \ll 1$, the bending energy dominates over the stretching energy and thus, all structures will adopt their spontaneous curvature, $R = R_0$. The situation will be similar to the growth of a crystal on a template of fixed curvature. In the bending-dominated regime, the free energy of formation of all these structures, when properly scaled by the characteristic elastic energy $4\pi R_0^2 Y$, only depends on two parameters: the scaled chemical potential $\Delta\tilde{\mu} \equiv \Delta\mu/(Y a_1)$ and the scaled line tension $\lambda \equiv \Lambda/(R_0 Y)$. Thus, it is possible to compare them and determine the most stable structure for a given set of conditions. The comparison is performed for different shapes having the same area $S$, that is having the same number of subunits.

The scaled free energy of formation of a hexagonally-ordered spherical cap of radius $R_0$ without defects made of a circular patch of radius $\rho_0$ (see *Figure 1a*) is

$$\Delta g_{cap} = -\frac{\Delta\tilde{\mu}}{4}x^2 + \frac{\lambda}{2}x + \frac{1}{1536}x^6 \tag{2}$$

where $x \equiv \rho_0 / R_0$ is the scaled patch size, and the third term is the in-plane elastic energy of a circular domain on a curved spherical surface (*Schneider and Gompper, 2007*; *Meng et al., 2014*; *Morozov and Bruinsma, 2010*). (*Equation 2* is an approximation strictly valid for small circular patches with an aperture angle $\theta \ll \pi$, since it is assumed that the perimeter of the shell is approximately the same as that of a circular disk, and a flat metric has been used to compute the in-plane elastic energy. However, we have found that a more accurate evaluation of the second and third terms in this equation [*Li et al., 2018*] does not alter significantly the main results.)

The stretching energy stored in the spherical shell grows fast with the area of the patch, and can be partially released by two different mechanisms: by the introduction of pentagonal defects (see *Figure 1b*), or by growing anisotropically forming curved ribbon-like crystalline domains (see *Figure 1c*).

The free energy of formation for a spherical cap with one defect is (*Morozov and Bruinsma, 2010*; *Castelnovo, 2017*)

$$\Delta g_{d_1} = \Delta g_{cap} + \frac{x^2}{1152}\left(1 - \frac{3}{2}x^2\right), \tag{3}$$

where the last term is the stretching energy due to a pentagonal disclination at the center of the cap. (The energy of an incomplete cap with one defect placed at an arbitrary location is calculated in *Li et al., 2018*. It is found that the Gaussian curvature attracts the disclination to the center of the cap while the defect self-energy pushes it towards the boundary. The net result is that the minimum energy corresponds to the defect located off the center of the cap. However, we have numerically verified that this approximation introduces only a very small error in our calculations for the scaled energy. This means that not noticeable effect is observed when the exact expression with the off-center defect is considered.) Such mechanism is energetically favorable only if the second term of *Equation 3* is negative, that is, if $x \geq \sqrt{2/3}$.

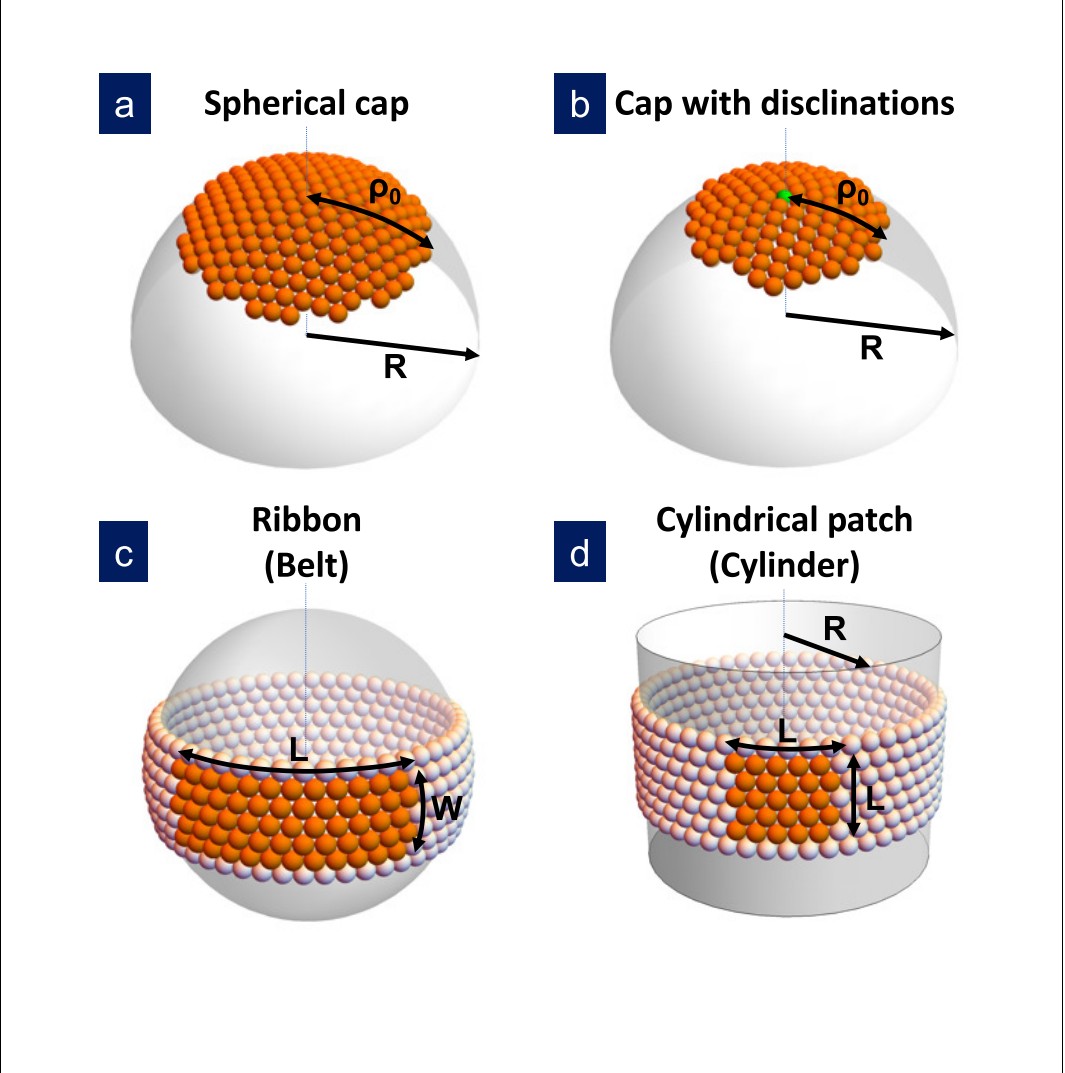

**Figure 1.** Sketch of the different structures considered in this study. (a) A hexagonally-ordered spherical cap of radius $R$ and geodesic radius $\rho_0$ without defects; (b) a spherical cap with a single disclination at the center (as shown) or multiple disclinations; (c) a rectangular ribbon with length $L$ and width $W$, that it is called *belt* when the length becomes $L = 2\pi R$; and (d) a cylindrical patch with size $L$, that eventually becomes a cylinder of radius $R$. In the bending-dominated regime, $R = R_0$.

For larger shells, the elastic strain is further released by the introduction of additional disclinations. The free energy of formation of a spherical shell with n 5-fold disclinations is (*Grason, 2012*; *Grason, 2016*; *Castelnovo, 2017*)

$$\Delta g_{d_n} = \Delta g_{cap} + g_{s_1} + g_{s_2} \qquad (4)$$

where $g_{s_1}$ is the self-energy of the isolated disclinations, and $g_{s_2}$ is their pairwise interaction, whose specific expressions are provided in the Appendix. When more than one defect appears, the minimum of the free energy typically occurs for a closed shell.

An alternative mechanism to alleviate stretching is the anisotropic growth of the originally spherical cap to adopt the shape of a defect-free rectangular curved stripe or ribbon. The free energy of formation of a ribbon of scaled length $l \equiv L/R_0$, width $w \equiv W/R_0$, and area $s = lw = \pi x^2$ growing on the surface of a sphere of radius $R_0$ is (*Schneider and Gompper, 2007*; *Majidi and Fearing, 2008*)

$$\Delta g_{rib} = -\frac{\Delta\tilde{\mu}}{4}x^2 + \frac{\lambda}{2}\frac{x^2}{w} + \frac{\lambda}{2\pi}w + \frac{9}{20480}x^2 w^4. \qquad (5)$$

Unlike the spherical cap, as the area of the patch increases, the ribbon grows longitudinally without limitation at a nearly fixed optimal width up to the point where $l = 2\pi$, where it forms a closed belt with energy

$$\Delta g_{belt} = -\frac{\Delta\tilde{\mu}}{4}x^2 + \lambda + \frac{9}{327680}x^{10}. \qquad (6)$$

The ribbon-like structure with the lowest energy is always a closed belt rather than the open ribbon, so we will focus our comparison with this structure.

Finally, an alternative to the curved belt could be a cylinder with one principal radius of curvature infinitely large and the other $R_0$ (see *Figure 1d*). The cylinder has the advantage of not having any in-plane stretching cost, but it has a bending energy penalty that prevents its formation in the bending-dominated limit (see the Appendix).

*Figure 2* shows a comparison of the energy landscape for the different structures for fixed values of $\Delta\tilde{\mu}$ and $\lambda$. The competition between the bulk energy gain, the line tension penalty and the stretching and bending costs will give rise, at the proper conditions, to a barrier that has to be overcome for triggering the formation of these structures. The height of this nucleation barrier and its location, corresponding to the critical cluster size, are mostly controlled by the bulk and line energy contributions, since the critical size typically occurs at small values of x. In terms of shell nucleation, the barrier for the formation of a spherical cap is always the smallest, since the line energy of a circular edge is always smaller than for a rectangular stripe of the same area. Accordingly, the initial embryo of all these structures will be a small spherical cap (*Paquay et al., 2017*). Neglecting the elastic terms, the critical size for the formation of a spherical shell will be $x^* \approx \lambda/\Delta\tilde{\mu}$, corresponding to a barrier height for nucleation of $\Delta g_{cap}^* \approx \lambda^2/(4\Delta\tilde{\mu})$. But rather than on the critical cluster for shell formation, we will be mostly interested in what is the most stable final structure for a given set of conditions.

Since the free energies of formation only depend on $\lambda$ and $\Delta\tilde{\mu}$ we can draw a universal phase diagram describing what is the structure (i.e. cap with or without defects, ribbon, or belt) with the lowest free energy in its stable size in terms of these two parameters. The term universal is intended to mean that the phase diagram is independent of the details of the capsomer-capsomer interactions such as range, preferred angle between capsomeres, bending rigidity, etc, as we corroborate with a coarse-grained simulation in the next section. *Figure 3a* shows the phase diagram in the bending-dominated limit, corresponding to $\gamma = 0$. As can be seen, belts are the most stable structure at low line tension $\lambda$ and chemical potential differences $\Delta\tilde{\mu}$. Closed shells with disclinations are the preferred structure for large values of $\Delta\tilde{\mu}$ or $\lambda$. The frontier between the belt zone and the cap with disclinations is approximately independent of $\lambda$ and located at $\Delta\tilde{\mu} \simeq 0.0020$. Additionally, a small triangular region where the most stable structure is a frustrated cap with only one disclination is also apparent. As shown in the Appendix, a stable defectless cap only appears as metastable structure, since it has always a larger energy than a belt, and it is thus non competitive as stable structure, even though it may have lower energies as intermediate in the assembly process.

## General case of arbitrary FvK number

Most small viral shells form without any underlying spherical template fixing their curvature. Therefore, it is very interesting to analyze shell formation at arbitrary FvK number, beyond the bending-dominated limit, and without the aid of an auxiliar template. In this general case, we have to consider the bending energy and the fact that the radius of the structures, $R$, may deviate from the spontaneous one, $R_0$, since it would be now dictated by the competition between stretching, bending, and rim energies. Using the expressions for the bending energy of a sphere and a cylinder of radius $R$ (see the Appendix), the free energy of formation of all structures analyzed in the previous section can be derived. Explicitly, the free energy of formation of a defectless spherical cap of radius $R$ becomes

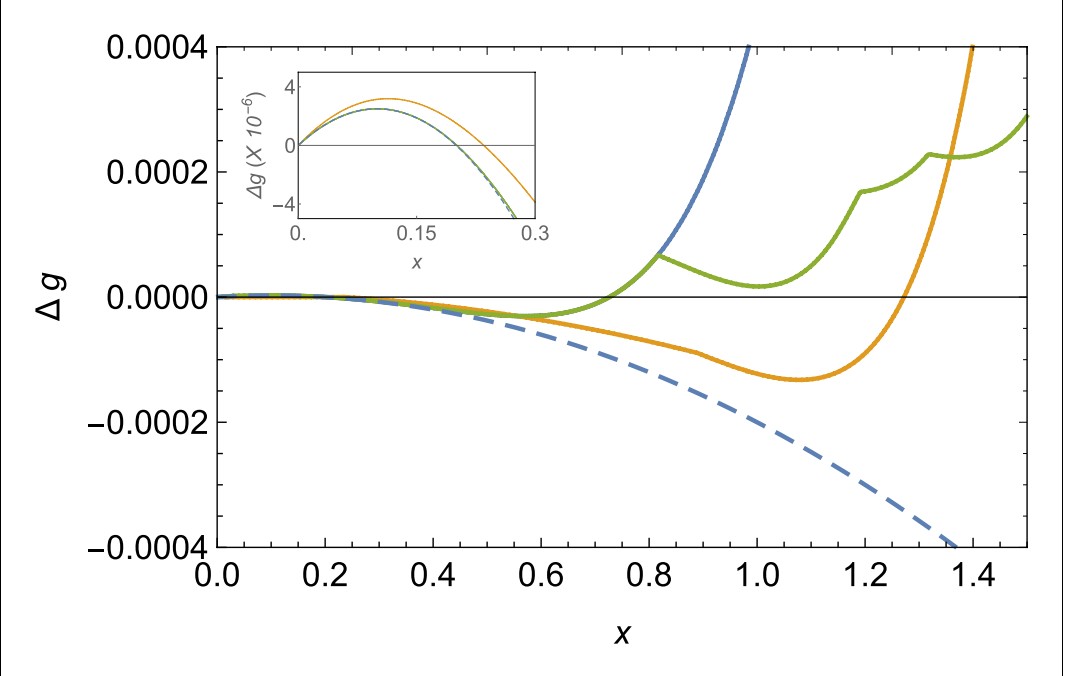

**Figure 2.** Comparison of free energy landscapes for different structures. Free energy of formation $\Delta g$ versus the radius of the patch x in the bending-dominated regime for a defectless spherical shell (blue line), a spherical shell with defects (green), and a ribbon/belt (orange) for $\lambda = 0.0001$ and $\Delta\tilde{\mu} = 0.001$. The optimal structure is the one with the minimum energy, which is the belt in this case. The dashed line represents the unfrustrated decrease of energy expected by the classical nucleation picture for the defectless spherical cap in the absence of elastic stresses. The inset zooms the nucleation barrier located at small patch sizes.

$$\Delta g_{cap}(\gamma) = -\frac{\Delta\tilde{\mu}}{4}x^2 + \frac{\lambda}{2}x + \frac{1}{1536}\frac{x^6}{r^4} + \frac{1}{4\gamma}x^2\left(\frac{1}{r}-1\right)^2, \tag{7}$$

where $r \equiv R/R_0$, and the optimal radius of the shell is given by

$$r^2(r-1) - \frac{\gamma}{192}x^4 = 0. \tag{8}$$

Deviations from the spontaneous radius (i.e $r = 1$) are only expected for large domain sizes or large FvK numbers.

As the domain size increases, it becomes more favorable to release the elastic stress by the introduction of one or many 5-fold disclinations. The free energy of formation of a spherical shell with one central defect is

$$\Delta g_{d_1}(\gamma) = \Delta g_{cap}(\gamma) + \frac{x^2}{1152}\left(1 - \frac{3}{2}\frac{x^2}{r^2}\right), \tag{9}$$

which becomes favorable over the defectless case when $x/r \geq \sqrt{2/3}$. The formation energy of a spherical shell with $n$-defects is

$$\Delta g_{d_n}(\gamma) = \Delta g_{cap}(\gamma) + g_{s_1}(r) + g_{s_2} \tag{10}$$

where the specific expressions for $g_{s_1}(r)$ and $g_{s_2}$ are written in the Appendix. Finally, the free energies of a closed belt and a cylinder (which are the ribbon-like and cylindrical-patch-like structures with the lowest energy) are

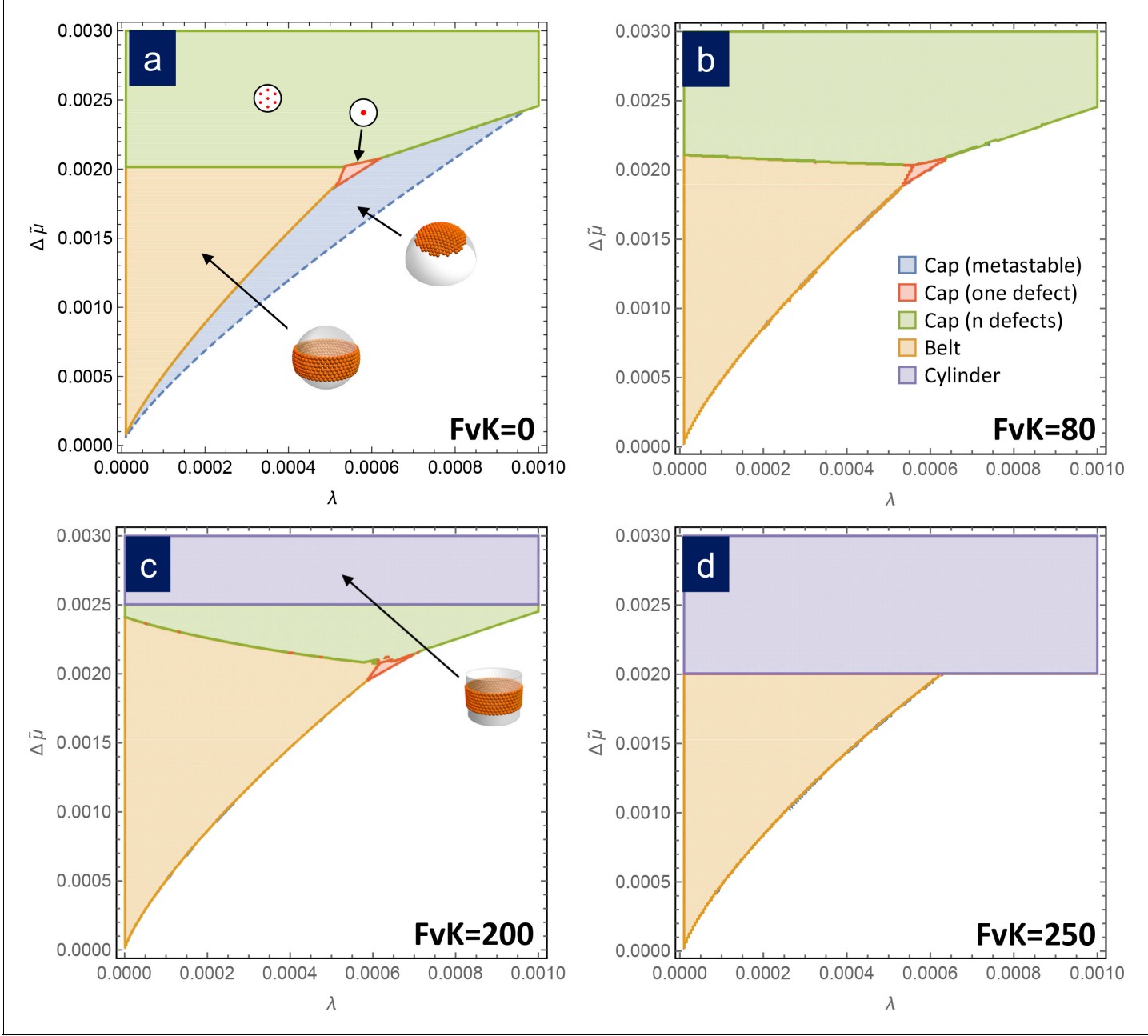

**Figure 3.** Assembly phase diagrams. Phase diagrams of the most stable structures in terms of the scaled chemical potential $\Delta\tilde{\mu}$ and the scaled line tension $\lambda$ for different values of the FvK number: a) $\gamma = 0$, corresponding to the bending-dominated regime, (b) $\gamma = 80$, (c) $\gamma = 200$, and d) $\gamma = 250$. Three possible equilibrium regions are present: belts (i.e. closed ribbons, in orange), frustrated capsids with one defect (red), closed shells with defects (green), and cylinders (purple). Additionally, a region corresponding to a metastable spherical cap without defects (blue) is shown only in (a). In the white region, the equilibrium state corresponds to disaggregated individual capsomers.

$$\Delta g_{belt}(\gamma) = -\frac{\Delta\tilde{\mu}}{4}x^2 + \lambda r + \frac{9}{327680}\frac{x^{10}}{r^8} + \frac{1}{4\gamma}x^2\left(\frac{1}{r}-1\right)^2,$$ (11)

and

$$\Delta g_{cyl}(\gamma) = -\frac{\Delta\tilde{\mu}}{4}x^2 + \lambda r + \frac{1}{8\gamma}x^2\left(1 + \left(\frac{1}{r}-1\right)^2\right), \tag{12}$$

respectively.

Remarkably, the free energy of formation of all these structures only depends on three scaled parameters: the chemical potential $\Delta\tilde{\mu}$, the line tension $\lambda$, and the FvK number $\gamma$. Thus, it is possible to compare them and draw a universal phase diagram for the most stable structure in terms of these three parameters, to contrast with the scenario for the bending-dominated limit. *Figure 3* shows phase diagrams for different values of the FvK number, showing the structure with the lowest free energy as a function of the normalized line tension $\lambda$ and chemical potential $\Delta\tilde{\mu}$. For small values of FvK, that is $\gamma \lesssim 100$, the phase diagram is essentially the same as in the bending-dominated case. As the FvK number increases, the region where belts are formed occupy a larger domain, while the region with closed caps with disclinations reduces its size. However, the most relevant change is the appearance of a zone at $\Delta\tilde{\mu} > 1/(2\gamma)$, where the cylinder is the optimal structure. This region progressively invades the other structures as the FvK number is increased. Roughly for $\gamma \simeq 250$ only cylinders and belts are expected to be stable structures. This is a very important result since it shows that spherical capsids cannot be self-assembled directly as stable structures at large FvK numbers.

The reason why cylinders dominate at large FvK numbers, corresponding to the regime where stretching dominates over bending, is because they have the advantage of not having any stretching energy cost (i.e. a flat sheet of hexamers can be bent into a cylinder without any stretching). A cylindrical structure having a radius equal to the spontaneous radius $R_0$, that is $r = 1$, will minimize the bending penalty and will have a free energy of formation, according to *Equation 12*, that decreases unboundedly with size when $\Delta\tilde{\mu} > 1/(2\gamma)$. In other words, once the formation of a cylinder becomes more favorable than free capsomers, it will continue growing without limit decreasing indefinitely its free energy of formation without paying any stretching cost, thus overcoming the energetic gain of any finite sized structure. This will be the case when $\Delta\tilde{\mu} > 1/(2\gamma)$. The larger the $\gamma$ (FvK), the smaller the $\Delta\tilde{\mu}$ required for this to occur and therefore, regions where finite sized structures where preferred start to be devoured by the region where cylinders dominate (purple regions in *Figure 3*). Making use of the definition of the scaled variables, the condition for the appearance of the cylindrical phase can be recast as $\Delta\mu \geq a_1\kappa/(2R_0^2)$. In other words, cylinders appear more easily (smaller $\Delta\mu$ required) for larger values of $R_0$, in agreement with previous results by *Castelnovo (2017)* predicting that cylinders should dominate for small spontaneous curvatures (large $R_0$).

## Simulation

A minimal coarse-grained model has been recently proposed to analyze the assembly of empty viral shells (*Aznar and Reguera, 2016*; *Aznar et al., 2018*) and other protein cages (*Garcia-Alles et al., 2017*). The model can successfully reproduce the assembly of the lowest spherical shell structures using capsomers, that is, pentamers and hexamers, as basic assembly units. Capsomers are coarse-grained at low resolution as effective spheres and their interaction is modelled using three contributions capturing the essential ingredients (see Materials and methods): a Mie-like potential describing the attraction driving the assembly and the excluded volume interaction between a pair of capsomers; an angular term accounting for the preferred orientation of the interaction between proteins; and a torsion term, included to distinguish the inner and outer surfaces of the capsomers, and to favor the formation of closed shells. The model has been implemented in a Brownian Dynamics simulation as described in Materials and methods.

One of the advantages of this simple model is that the parameters of the interaction can be related to the elastic constants (*Aznar and Reguera, 2016*) (see Materials and methods). In terms of these, the three relevant parameters controlling the assembly become $\gamma \equiv \frac{YR_0^2}{\kappa} = \frac{4nm\alpha^2}{9\cos^2\nu'}$, $\lambda \equiv \frac{\Lambda}{YR_0} = \frac{2\cos\nu}{nm}$, and $\Delta\tilde{\mu} \equiv \frac{\Delta\mu}{a_1 Y} = \frac{2\sqrt{3}}{\pi nm}\frac{\Delta\mu}{\epsilon_0} = \frac{2\sqrt{3}}{\pi nm}\frac{k_B T \ln c_1/c^*}{\epsilon_0}$.

Thus, by changing the parameters of the model (mainly the exponents $n$ and $m$ controlling the range of the interaction, the preferred angle of interaction between capsomers $\nu$, the local bending rigitidy $\alpha$, and the concentration $c_1$ which controls the effective chemical potential $\Delta\mu$) we can explore the universality and the different scenarios of assembly discussed in the previous section.

*Figure 4* shows the results of simulations using different sets of parameters represented in scaled units and contrasted with the theoretical phase diagram for $\gamma = 80$. For $\lambda = 0.00084$, that is a relatively large line tension, at low concentration of capsomers, the seed dissolves and no nucleation or growth occurs. As the capsomer concentration is progressively increased, a metastable defectless shell and a closed spherical shell with typically 12 defects form, as expected by the theory. At very high concentrations, nucleation occurs simultaneously at many sites, and the final outcome of the

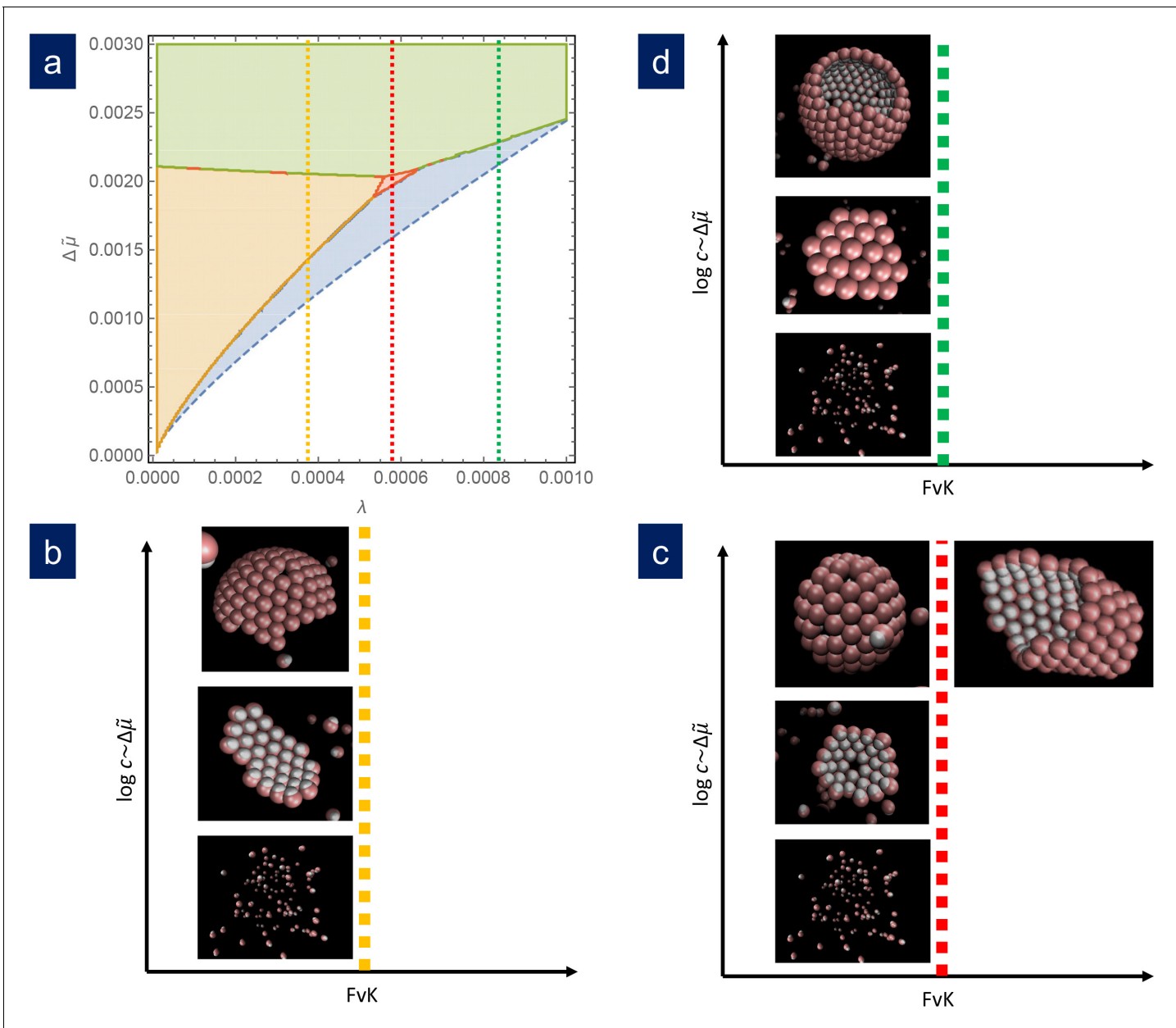

**Figure 4.** Comparison of simulation results with the theoretical phase diagram. (a) Phase diagram of the most stable structure in terms of the scaled chemical potential $\Delta\tilde{\mu}$ and the scaled line tension $\lambda$ for $\gamma = 80$. Snapshots of the final outcome of the simulation for different initial capsomer concentrations for: b) $\lambda = 0.00037$, obtained with $\nu = 1.45$ and a $m = 36$, $n = 18$ potential. By increasing the concentration of capsomers one goes from a dissassembled state to the formation of a ribbon to a spherical shell with defects. (The ribbon and spherical shell are not closed in the snapshots due to the limited number of capsomers and finite simulation time). (c) $\lambda = 0.000525$, obtained with $\nu = 1.38$ and a $m = 36$, $n = 18$ potential. As concentration increases, one goes from a shell with one defect to a complete shell with many defects. By increasing the FvK number to $\gamma = 1280$ (by setting $\alpha = 0.4$), the simulations clearly forms cylindrical tubes. (d) $\lambda = 0.00084$, obtained with $\nu = 1.45$ and a $m = 24$, $n = 12$ potential. In this case, the sequence is: disagreggated, metastable defectless shell, and closed spherical shell with many defects (a partial shell is shown). The simulation results agree qualitatively with the predictions from the theoretical phase diagram.

simulation are many fragments of spherical capsids that cannot grow any further due to the depletion of free capsomers in solution. That would correspond to kinetic trapping, which is an interesting alternative mechanism to prevent the correct capsid assembly, that will be analyzed in a future work. For $\lambda = 0.000585$, as the concentration is increased we obtained the expected sequence of: seed disolution; formation of a stable cap with a single 5-fold defect in a very narrow range of concentrations; and the formation of closed caps with many defects. Finally for $\lambda = 0.000372$, that is a relatively small line tension, as the capsomer concentration is increased we go from no assembly, to the formation of ribbon-like stripes, to the growth of spherical shells with many defects. As naively expected, as $\lambda$ increases, higher scaled chemical potentials are needed to nucleate the structures. Finally, by increasing the parameter $\alpha$, the bending rigidity is reduced and assembly at higher FvK can be analyzed. The results of the simulations show that as the FvK number is increased, the formation of spherical shells is overriden by the formation of cylindrical bodies, as shown in *Figure 4c*, that also competes with other elongated structures such as spherocylinders or even conical shapes (see *Appendix 1—figure 3*). Remarkably, simulations that have been performed for widely different values of the interaction parameters, when properly scaled, all fall into the predicted picture. Therefore, the simulation results nicely confirm almost quantitatively the universality of the fate of the assembly and the potential scenarios discussed in the theory. A precise quantitative comparison between the theory and the simulations has not been performed, since they are done at slightly different conditions. While theory assumes a reservoir of capsomers, the simulations are done at fixed total number of subunits. This implies that, as the assembly proceeds, the concentration of the remaining free particles, and consequently the chemical potential, decreases. For this reason, we have not intended to reproduce with precision the borders of the phase diagram using the simulations. The fact that the chemical potential is not strictly constant in the simulations due to the depletion of free subunits may cause quantitative discrepancies when comparing with theory, but does not alter the relative stability of the different shapes analyzed.

## Discussion

We have provided a comprehensive analysis of non-templated assembly of curved elastic shells, taking into account all relevant ingredients (i.e. chemical potential, line tension, spontaneous curvature, and elastic contributions) and the potential formation of non-spherical shapes. The importance of accounting for all these ingredients becomes evident, for instance, in the study of the stability of the defectless spherical cap, which turns out to be always metastable, its global stability hindered by the introduction of defects (at high line tensions) or the formation of ribbons (at low line tensions). Our analysis also shows that the outcome of the assembly not only depends on elastic considerations, but also on the assembly conditions, represented here by the scaled chemical potential. Hence, either belts or closed spherical shells or cylinders may be obtained as the most stable structure for fixed interaction parameters, depending on the concentration of assembly units. When assembly takes place at conditions near the vicinity of a phase boundary, a mixture of the two phases, or a structure resulting from their combination (e.g. a spherocylinder) may form. This may justify the observation of coexisting tubes and spherical capsids in the in vitro assembly of viruses such as SV40 (*Kanesashi et al., 2003*).

Although, for the sake of simplicity, our theoretical analysis has been performed using the continuous and small curvature approximations, we have verified that releasing these approximations does not alter significantly the results. The exact expression of the perimeter of the growing edge (*Zandi et al., 2006*; *Gómez et al., 2015*) influences the height and location of the nucleation barrier, but has a minor impact on the properties of the final stable structure. The accurate evaluation of the in-plane elastic cost of defects taking into account their spatial distribution (*Li et al., 2018*), modifies the energies of the growing shell, but does not modify significantly the stability of the final structure.

Simulations of a coarse-grained model made using widely different values for the parameters and interaction range confirm that the outcome of the assembly only depends on three scaled parameters: the scaled chemical potential $\Delta\tilde{\mu}$, line tension $\lambda$, and FvK number $\gamma$. Thus, the assembly phase diagram is universal, and different protein shells, interaction potentials and coarse-grained models can be recast into a unifying picture of assembly, that could guide the efficient production of artificial viral cages. For instance, our analysis indicates that relatively long-range interactions are desirable to increase the line tension, decrease the FvK number and facilitate the assembly of closed

spherical shells. In fact, spherical shells with icosahedral symmetry and triangulation number T > 7 could be successfully assembled in simulations without any template or scaffolding protein, provided that the line tension and FvK number are adequate. Alternatively, chemical or physical modifications that increase the FvK number or reduce the line tension or the effective concentration may become a potential therapeutic target to prevent viral replication by inducing the formation of open, and presumably non-infective, cylindrical or belt-like structures. Experimentally, the chemical potential can be tuned by the total protein concentration or by the addition of crowding agents. The line tension (which depends on the strength of the binding interaction), could be modified by the temperature, the pH and the salt concentration. The bending rigidity and spontaneous radius of curvature are also presumably controlled by pH and the presence, concentration and nature of ions or auxiliary proteins in solution. Further experimental and theoretical investigations are required to make a precise quantitative connection between the physical parameters controlling the assembly and experiments.

Triggering the formation of closed spherical shells with an incorrect radius, triangulation number (*Caspar and Klug, 1962*), or arrangement of proteins could also be an alternative to interfere with the assembly of the right viral capsid. But in our study, we have focused on mechanisms interferring with the closing of the shell by elastic frustration, rather than classifying the specific radius and triangulation number of the resulting spherical structure. In addition, we do not consider the situation in which the capsomers interact with cargo. Such interactions are crucial for viruses that co-assemble with their genetic material or a cargo, but this is beyond the scope of the present study.

A very important conclusion of our analysis is that spherical capsids cannot be self-assembled directly as stable structures at large FvK numbers. This may explain why some viruses that require high mechanical resistance, for instance many dsDNA bacteriophages such as lambda, HK97 and P22, first assemble a relatively soft spherical procapsid before suffering a maturation transition (*Roos et al., 2012*; *Johnson, 2010*) that flattens out their faces, which is a clear signature of a high FvK number (*Lidmar et al., 2003*). The results of our work indicate that a one-step assembly of a spherical shell with the high elastic resistance and Fvk number of the final structure is not viable. *Table 1* compares the estimated elastic properties of different empty capsids of real viruses. The table clearly shows that viruses like CCMV or SV40 that assemble easily in vitro as spherical shells, have estimated values of the scaled line tension and FvK in the region where these structures are expected to be stable outcomes of the assembly. Contrarily, the high FvK number of the mature bacteriophage lambda will prevent its direct assembly. However, its procapsid, which is the first structure that is assembled, has a larger scaled line tension and smaller FvK that would facilitate a successful assembly. (The FvK number of lambda procapsid listed in *Table 1* is probably overestimated, given its noticeable spherical shell. In addition, we have found in our simulations that even though the theoretical threshold for the disappearance of spherical shells as stable structures is around $\gamma = 250$, in practice larger FvK numbers are typically required to obtain cylindrical structures since the nucleation barrier for their formation is larger than for the metastable spherical shell).

**Table 1.** Estimates of the main geometric and elastic properties of different non-enveloped empty viral capsids.
The Young's modulus $E$ has been evaluated from AFM nanoindentation experiments (*Mateu, 2012*; *Michel et al., 2006*; *Ivanovska et al., 2007*; *Sae-Ueng et al., 2014*) and, for SV40, from the experimental spring constant (*van Rosmalen et al., 2018*) using the standard thin shell formula $k = 2.25Eh^2/R$ (*Ivanovska et al., 2004*). The 2D Young's Modulus was calculated as $Y = Eh$; the effective diameter of the capsomers as (*Santolaria, 2011*) $\sigma = R/\sqrt{\frac{5\sqrt{3}}{\pi}\left(T + \frac{1}{\sqrt{3}}cot\left(\frac{\pi}{5}\right) - 1\right)}$, where $T$ is the triangulation number; the line tension as (*Luque et al., 2012*) $\lambda = \frac{2\epsilon_0}{\sqrt{3}\sigma}$ considering a typical binding energy $\epsilon_0 \approx 10k_BT$; and the FvK number as $\gamma = 12(1 - \nu_p^2)(R/h)^2$, with $\nu_p = 0.3$ (*Buenemann and Lenz, 2008*).

| Virus | T-number | Diameter (nm) | Thickness h (nm) | E (Gpa) | Y (N/m) | σ (nm) | Scaled line tension λ | Föppl-von Karman γ |
|---|---|---|---|---|---|---|---|---|
| CCMV | 3 | 28 | 3.8 | 0.14 | 0.53 | 5.9 | 0.00107 | 148 |
| λ Procapsid | 7 | 50 | 4.0 | 0.16 | 0.64 | 6.8 | 0.000436 | 427 |
| λ Capsid | 7 | 63 | 1.8 | 1.0 | 1.8 | 8.6 | 0.0000976 | 3344 |
| SV40 | 7 | 45 | 6.0 | 0.033 | 0.2 | 6.1 | 0.00174 | 152 |

In summary, we have seen that the fate of the assembly is controlled by a universal phase diagram in terms of three scaled parameters: line tension, chemical potential and FvK number. The phase diagrams shed light on the physics controlling the assembly of curved shells, and could guide assembly experiments to achieve either an efficient assembly of artificial viral shells of desired geometry and mechanical properties or, alternatively, to envisage the conditions needed to impede viral infections by arresting viral assembly or inducing missasembly into a non-infective structure.

# Materials and methods

## Coarse-grained model and simulation details

The simulation model, introduced in *Aznar and Reguera (2016)*; *Aznar et al. (2018)*, is coarse-grained at the level of capsomers which are represented as effective spheres of two different diameters: $\sigma_h$ and $\sigma_p$, reflecting the fact that hexamers and pentamers are made of a different number of proteins (six and five, respectively). The interaction between capsomers, $V = V_{LJ} \cdot V_a \cdot V_{tor}$, is modeled using three contributions: a Mie-like, an angular, and a torsion potential. The Mie-like potential

$$V_{LJ}(\mathbf{r}_{ij}) = \epsilon_{ij} \frac{n}{m-n} \left[ \left( \frac{\sigma_{ij}}{r} \right)^m - \frac{m}{n} \left( \frac{\sigma_{ij}}{r} \right)^n \right] , \tag{13}$$

describes the binding and the excluded volume interaction between a pair of capsomers in terms of their relative distance, $\mathbf{r}_{ij}$ is the equilibrium distance corresponding to the minimum of the potential, r is the distance between capsomers centers, $\epsilon_{ij}$ is the binding energy between capsomers, and m and n represent the power of the repulsive and attractive interaction terms, respectively, which set the range of the interaction potential. The angular contribution is given by

$$V_a(\mathbf{r}_{ij}, \Omega_i, \Omega_j) = \exp\left( -\frac{(\theta_{ij} - \nu)^2}{2\alpha^2} \right) \exp\left( -\frac{(\theta_{ji} - \nu)^2}{2\alpha^2} \right) , \tag{14}$$

where $\theta_{ij}$ is the angle between the vector $\Omega_i$, describing the spatial orientation of the capsomer, and the unit vector $\mathbf{r}_{ij}$. The parameter $\nu$ is the preferred angle of interaction between proteins of different capsomers, and the parameter $\alpha$ controls the local bending stiffness, that is, the energy cost required to bend two capsomers out of their preferred angle of interaction. The torsion term is given by

$$V_{tor}(\Omega_i, \Omega_j) = \exp\left( -k_t \frac{(1 - \cos\xi)}{2} \right) , \tag{15}$$

where $k_t$ is the torsion constant and $\xi$ is the angle between the planes defined by the unit vector $\mathbf{r}_{ij}$ and both orientation vectors.

The elastic properties of a shell can be related to the main parameters of the interaction. In particular the Young's modulus is approximately given by $Y = \frac{2nm}{\sqrt{3}} \frac{\epsilon_0}{\sigma^2}$, the bending rigidity is $\kappa = \frac{3\sqrt{3}}{8} \frac{\epsilon_0}{\alpha^2}$, and the preferred radius of curvature is $R_0 = -\frac{\sigma}{2\cos\nu}$. The line tension of a partially formed cap can be approximated by *Luque et al. (2012)* $\lambda = \frac{2\epsilon_0}{\sqrt{3}\sigma}$, and the chemical potential difference that controls the assembly is given by $\Delta\mu = k_B T \ln(c_1/c^*)$, where $c_1$ is the concentration of free capsomers and $c^*$ is the critical concentration (*Zandi et al., 2006*).

The model has been implemented in a Brownian Dynamics simulation code using a simple stochastic Euler's integration algorithm, as described in *Aznar et al. (2018)*. Simulations were made with only one type of capsomers. We worked using reduced units in terms of the diameter of the basic building blocks $\sigma$, their diffusion coefficient $D$, and the binding energy $\epsilon_0$. In these reduced units, the parameters used in the simulation are: torsion constant $k_t = 1.5$, reduced temperature $T = 0.1$, corresponding to a binding energy between capsomers of $10 k_B T$, representing the typical order of magnitude of the strength of interactions between viral capsid proteins.

Since, in all cases, the critical nucleus is a partial spherical cap, all simulations were started using a small spherical cap of 19 units with the spontaneous curvature as initial seed. The remaining capsomers up to a total of $N = 200 - 400$ were initially placed randomly inside a cubic box with periodic boundary conditions. The simulations run for a total of $2 \times 10^9$ steps and the final structures were

analyzed. To verify the universality of the phase diagram, we performed an extensive set of simulations with different interaction parameters. More specifically, the interaction range was varied from $m = 12, n = 6$ to $m = 48, n = 24$; the spontaneous angle in the range $1.24 < \nu < 1.45$; the bending stiffness in the range $0.05 < \alpha < 0.4$; and the concentration of capsomers was varied from $\rho = 0.005$ to $\rho = 0.05$.

## Acknowledgements

CIM appreciates the hospitality of Dr. David Reguera and the Departament de Física de la Matèria Condensada of the Universitat de Barcelona where this work was carried out during a sabbatical leave and subsequent visits. CIM received financial support provided by DGAPA-UNAM through a Sabbatical Fellowship and by grants DGAPA IN-110516 and IN-103419. DR acknowledges funding from the Spanish government through grants FIS2015-67837-P and PGC2018-098373-B-I00 (MINECO/FEDER, UE).

## Additional information

### Funding

| Funder | Grant reference number | Author |
|---|---|---|
| Universidad Nacional Autónoma de México | DGAPA IN-110516 | Carlos I Mendoza |
| Universidad Nacional Autónoma de México | DGAPA IN-103419 | Carlos I Mendoza |
| Gobierno de Espana | FIS2015-67837-P | David Reguera |
| Ministerio de Economía y Competitividad | PGC2018-098373-B-I00 | David Reguera |
| European Regional Development Fund | PGC2018-098373-B-I00 | David Reguera |
| Universidad Nacional Autónoma de México | Sabbatical Fellowship | Carlos I Mendoza |

The funders had no role in study design, data collection and interpretation, or the decision to submit the work for publication.

### Author contributions

Carlos I Mendoza, David Reguera, Conceptualization, Formal analysis, Funding acquisition, Investigation, Methodology

### Author ORCIDs

Carlos I Mendoza https://orcid.org/0000-0001-9769-240X
David Reguera http://orcid.org/0000-0001-6395-6112

### Decision letter and Author response

Decision letter https://doi.org/10.7554/eLife.52525.sa1
Author response https://doi.org/10.7554/eLife.52525.sa2

## Additional files

### Supplementary files
• Transparent reporting form

### Data availability

All data generated or analysed during this study are included in the manuscript and supporting files.

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

## Appendix 1

# Elastic Model of Shell Assembly

The elastic energy in the formation of a curved shell has two contributions: the bending energy, $G_b$, associated with deviations from the spontaneous curvature, and the in-plane elastic energy, $G_s$. The bending energy is described using the expression (*Castelnovo, 2017*):

$$G_b = \frac{\kappa}{2} \int \left[ \left( H - \frac{2}{R_0} \right)^2 - 2K - \frac{2}{R_0} \left( \frac{1}{R_0} - H \right) \right] dS \tag{A1}$$

where, H is twice the mean curvature of the shell, $R_0$ is the spontaneous radius of curvature, K is the Gaussian curvature and $\kappa$ is the bending modulus.

The expression for the in-plane elastic energy of the different analyzed structures is based on previous results from continuum elasticity theory for the deformation of thin plates (*Seung and Nelson, 1988*; *Lidmar et al., 2003*; *Majidi and Fearing, 2008*; *Morozov and Bruinsma, 2010*; *Meng et al., 2014*; *Grason, 2016*; *Paquay et al., 2017*; *Castelnovo, 2017*; *Schneider and Gompper, 2007*). The structures analized are: a hexagonally-ordered spherical caps without defects; a spherical cap with one central defect; a spherical cap with n defects; a ribbon; a belt; a cylindrical patch; and a cylinder. The Föppl-von Kármán number (FvK), defined in this work as $\gamma \equiv YR_0^2/\kappa$, dictates the relative importance of bending and stretching contributions.

# Bending-dominated regime

For $\gamma \ll 1$, the bending term dominates and forces all structures to adopt the spontaneous curvature $R = R_0$. We will derive the free energy of formation of the different structures in the small curvature approximation, and compare their relative stability under assembly conditions. The comparison is performed for different shapes having the same area S, that is having the same number of subunits.

The in-plane elastic energy of a circular domain of geodesic radius $\rho_0$ on a curved spherical surface of radius R is given by *Schneider and Gompper (2007)*; *Morozov and Bruinsma, 2010*; *Meng et al. (2014)*

$$\Delta G_s^{cap} = \frac{\pi Y}{384} \frac{\rho_0^6}{R^4}. \tag{A2}$$

Accordingly, its free energy of formation becomes

$$\Delta G_{cap} = -\frac{\pi \rho_0^2}{a_1} \Delta\mu + 2\pi\rho_0\lambda + \frac{\pi Y}{384} \frac{\rho_0^6}{R^4}, \tag{A3}$$

or in scaled units

$$\Delta g_{cap} = -\frac{\Delta\tilde{\mu}}{4} x^2 + \frac{\lambda}{2} x + \frac{1}{1536} x^6, \tag{A4}$$

where $\Delta g_{cap} \equiv \frac{\Delta G_{cap}}{4\pi R_0^2 Y}$ is the free energy of formation divided by the characteristic elastic energy $4\pi R_0^2 Y$, $x \equiv \rho_0/R_0$ is the scaled patch radius, $\lambda \equiv \Lambda/(R_0 Y)$ is the scaled line tension, and $\Delta\tilde{\mu} \equiv \Delta\mu/(Ya_1)$ is the scaled chemical potential. The bulk energy grows as $x^2$, the rim energy as x, and the elastic stress as $x^6$. The competition between these three contributions determines the shape of the $\Delta g_{cap}(x)$ curve (see *Appendix 1—figure 1*). For $\Delta\tilde{\mu}$ small, the positive second and third terms of *Equation A4* dominate, thus, grows monotonically as shown in *Appendix 1—figure 1*. However, as $\Delta\tilde{\mu}$ increases, there is a particular value

$$\Delta\tilde{\mu}^{ms} = \frac{5}{8}\lambda^{4/5}, \tag{A5}$$

obtained by setting $\frac{d\Delta g_{cap}}{dx} = 0$ and $\frac{d^2\Delta g_{cap}}{dx^2} = 0$, at which an inflection point located at

$$x^{ms} = 2\lambda^{1/5} \tag{A6}$$

appears. For $\Delta\tilde{\mu} > \Delta\tilde{\mu}^{ms}$, the free energy landscape has a maximum, signaling the nucleation barrier, but also a local minimum, $x_{min}$, corresponding to a locally-stable spherical cap. Thus, unlike in the standard case where beyond the critical size the free energy goes steadily down and the shell can grow until closing, the high elastic cost associated with the curvature of the shell will prevent its further growth and force it to reach an equilibrium size (**Grason, 2016**). The condition $\Delta\tilde{\mu} \geq \Delta\tilde{\mu}^{ms}$ marks the onset of a metastable region where the value of free energy $\Delta g_{cap}(x_{min})$ of the locally-stable shell is larger than its value for the dissembled state at x = 0. A fully stable partial shell is obtained if $\Delta g_{cap}(x_{min}) \leq 0$ at the minimum. The onset of the stable region can be obtained from the conditions $\Delta g_{cap} = 0$ and $d\Delta g_{cap}/dx = 0$, which allows to find the critical value

$$\Delta\tilde{\mu}^s_{cap} = \frac{5}{4}\left(\frac{\lambda^4}{6}\right)^{1/5}, \tag{A7}$$

beyond which $\Delta g_{cap}(x_{min}) \leq 0$. The corresponding value of x is

$$x^s = 2(6\lambda)^{1/5}, \tag{A8}$$

and represents the minimum size of a partial spherical shell without defects to be stable. The size of stable partial shells grows as $\Delta\tilde{\mu}$ increases. For large patches or when $\gamma \ll 1$, the rim energy can be neglected and the location of the stable cap size is described approximately by $x_{eq} \approx (128\Delta\tilde{\mu})^{1/4}$.

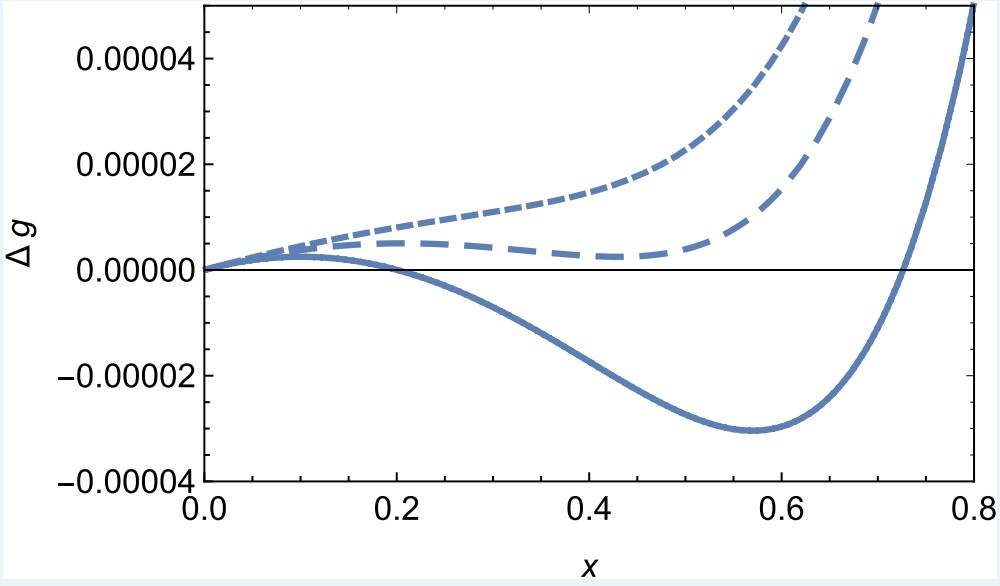

**Appendix 1—figure 1.** Free energy of formation $\Delta g_{cap}$ of a spherical cap without defects, *Equation A4*, versus the radius of the patch $x$ for $\lambda = 0.0001$ and different values of the scaled chemical potential $\Delta\tilde{\mu}$, illustrating the situations in which no assembly is possible (short dashed line, $\Delta\tilde{\mu} = 0.0002$), a geometrically frustrated metastastable cap (dashed line, $\Delta\tilde{\mu} = 0.0005$) or stable (solid line, $\Delta\tilde{\mu} = 0.001$) finite shell.

When the area of the spherical crystalline patch gets large, it becomes favorable to introduce defects to release the elastic stress. The in-plane elastic energy of a spherical cap with one 5-fold disclination is (*Morozov and Bruinsma, 2010*; *Grason, 2016*)

$$\Delta G_{d_1}^s = \frac{\pi Y \rho_0^2}{288} \left[ 1 - \frac{3}{2} \frac{\rho_0^2}{R_0^2} \right].$$

(A9)

Thus, the free energy of formation, in scaled units, for a spherical cap with one defect at the center becomes

$$\Delta g_{d_1} = \Delta g_{cap} + \frac{x^2}{1152} \left( 1 - \frac{3}{2} x^2 \right).$$

(A10)

Such mechanism is energetically favorable only if the second term of *Equation A10* is negative, that is, if $x \geq \sqrt{2/3}$. The fact that for $x \geq \sqrt{2/3}$ a shell with a defect becomes energetically favorable imposes an important restriction on the values that the scaled chemical potential and line tension may have in order to allow the existence of a stable defectless spherical cap. Imposing $x^s \leq \sqrt{2/3}$ one gets the requirements $\lambda \leq 6^{7/2}$ and $\Delta\tilde{\mu} \leq 5/864$.

Using the conditions $\Delta g_{d_1} = 0$ and $d\Delta g_{d_1}/dx = 0$, we can find the onset of stability for a circular cap with one central disclination, described by

$$\Delta\tilde{\mu}_{d_1}^s = \frac{2\lambda}{x_1} + \frac{4 - 6x_1^2 + 3x_1^4}{1152},$$

(A11)

with

$$x_1 = \left( \frac{1 + \sqrt{1 + 768\lambda}}{2} \right)^{1/2}.$$

(A12)

For larger shells, the elastic strain is further released by the introduction of additional disclinations. The resulting free energy of formation of a spherical shell with $n$ 5-fold disclinations in scaled units is (*Grason, 2012*; *Grason, 2016*; *Castelnovo, 2017*)

$$\Delta g_{d_n} = \Delta g_{cap} + g_{s_1} + g_{s_2}$$

(A13)

where

$$g_{s_1} = \frac{x^2}{1152} \left( 1 - \frac{3}{2} x^2 \right) \sum_{\alpha=1}^{n} \left( 1 - \frac{x_\alpha^2}{x^2} \right)^2,$$

(A14)

is the self-energy of the isolated disclinations, and

$$g_{s_2} = \frac{x^2}{1152} \sum_{\beta \neq \alpha} V_{int}(\mathbf{x}_\alpha, \mathbf{x}_\beta),$$

(A15)

is the pairwise interaction of disclinations, with

$$V_{int}(\mathbf{x}_\alpha, \mathbf{x}_\beta) = \left\{ \left( 1 - \frac{x_\alpha^2}{x^2} \right) \left( 1 - \frac{x_\beta^2}{x^2} \right) + \frac{|\mathbf{x}_\alpha - \mathbf{x}_\beta|^2}{x^2} \ln \left[ \frac{|\mathbf{x}_\alpha - \mathbf{x}_\beta|^2}{\left( x^2 - x_\alpha^2 \right) \left( x^2 - x_\beta^2 \right)/x^2 + |\mathbf{x}_\alpha - \mathbf{x}_\beta|^2} \right] \right\},$$

(A16)

being $x_\alpha \equiv \rho_\alpha/\rho_0$, $\rho_\alpha$ the geodesical position of the disclinations and $|\mathbf{x}_\alpha - \mathbf{x}_\beta|$ is the (normalized) geodesic distance between disclination $\alpha$ and $\beta$.

When more than one defect appear, the minimum of the free energy typically occurs for a closed shell, corresponding in the small curvature approximation to $x \simeq 2$ in which case, the stability region appears for $\Delta\tilde{\mu}$ larger than

$$\Delta\tilde{\mu}_{d_n}^s \simeq \lambda + 0.00146.$$

(A17)

and the cap consists of a fully closed shell with defects.

The free energy of formation of a ribbon of length $L$ and width $W$, growing on the surface of a sphere of radius $R_0$ is (**Schneider and Gompper, 2007**; **Majidi and Fearing, 2008**)

$$\Delta G_{rib} = -\frac{LW}{a_1}\Delta\mu + 2\Lambda(L+W) + \frac{9Y}{5120}\frac{W^5 L}{R_0^4}. \tag{A18}$$

The energetic advantage of this configuration over the spherical cap is that, for a fixed width $W$, the in-plane elastic energy only grows linearly with length. In order to compare the energy of ribbons made with similar number of units as the spherical cap, we consider that both structures have the same area. That is, $S = \pi\rho_0^2 = LW$. Thus, this energy can be rewritten in dimensionless terms as

$$\Delta g_{rib} = -\frac{\Delta\tilde{\mu}}{4}x^2 + \frac{\lambda}{2}\frac{x^2}{w} + \frac{\lambda}{2\pi}w + \frac{9}{20480}x^2 w^4 \tag{A19}$$

where $w \equiv W/R_0$. The optimal width of the ribbon is obtained by minimization that is, from $\partial\Delta g_{rib}/\partial w = 0$, yielding

$$-\lambda\frac{x^2}{w^2} + \frac{\lambda}{\pi} + \frac{9}{2560}x^2 w^3 = 0. \tag{A20}$$

As the area of the patch increases, the ribbon grows longitudinally at a nearly fixed optimal width up to the point where $l = 2\pi$, where it forms a closed belt with energy

$$\Delta g_{belt} = -\frac{\Delta\tilde{\mu}}{4}x^2 + \lambda + \frac{9}{327680}x^{10}. \tag{A21}$$

The elastic energy of the belt grows very steeply with the patch size, since after closing the ribbon, it can only grow by increasing its width at a large stretching cost. It can be shown that the equilibrium ribbon-like structure with the lowest energy is always a closed belt rather than the open ribbon. So the competing structures are the spherical cap with or without defects and the belt. The minimum in the free energy, corresponding to a stable belt, is located at

$$x_{belt}^s = 2\left(\frac{64\Delta\tilde{\mu}}{9}\right)^{1/8}, \tag{A22}$$

and the stability region for the belt occurs when $\Delta\tilde{\mu}$ is larger than

$$\Delta\tilde{\mu}_{belt}^s = \left(\frac{5}{8}\sqrt{\frac{3}{2}}\lambda\right)^{4/5}. \tag{A23}$$

Since $\Delta\tilde{\mu}_{belt}^s < \Delta\tilde{\mu}_{cap}^s$, that leads to the important result that closed belts become stable before defectless spherical caps. Therefore, defectless spherical caps can only be at most metastable structures.

Finally, an alternative to the curved ribbon is a cylindrical stripe of scaled width $w$ and length $l$ having one of his principal radius of curvature zero and a scaled energy

$$\Delta g_{stripe} = -\frac{\Delta\tilde{\mu}}{4}x^2 + \frac{\lambda}{2}\frac{x^2}{w} + \frac{\lambda}{2\pi}w + \frac{1}{8\gamma}x^2 \tag{A24}$$

As in the case of the ribbon and the belt, the cylindrical patch eventually closes when $l = 2\pi r$ into a cylinder whose energy of formation is

$$\Delta g_{cyl} = \frac{1}{4}x^2\left(-\Delta\tilde{\mu} + \frac{1}{2\gamma}\right) + \lambda. \tag{A25}$$

The cylinder has the advantage of not having any in-plane stretching cost, but it has a bending energy penalty described by the second term inside the parentesis. An energetically favorable cylinder requires $\Delta\tilde{\mu} > 1/(2\gamma)$ for its formation, meaning that cylindrical shells cannot be formed in the bending-dominated limit, corresponding to $\gamma \to 0$.

## General case of arbitrary FvK

The description of the free energy of formation of shells at arbitrary FvK numbers involves the consideration of the bending energy and of the radius of the structures as an additional free parameter that may now deviate from the spontatneous radius $R_0$. Particularizing *Equation A1*, the bending energy of a sphere of radius $R$ in scaled units is

$$\Delta g_b^{sph} = \frac{1}{4\gamma}x^2\left(\frac{1}{r}-1\right)^2,$$ (A26)

while for a cylinder of radius $R$ reads

$$\Delta g_b^{cyl} = \frac{1}{8\gamma}x^2\left(1+\left(\frac{1}{r}-1\right)^2\right)$$ (A27)

where $r \equiv R/R_0$ and the Föppl-von Kármán number (FvK), quantifying the ratio of stretching and bending energies, is still defined as $\gamma \equiv YR_0^2/\kappa$.

Using these expressions we can generalize the free energy of formation of all structures analyzed in the previous section. Explicitly, in reduced units, the free energy of formation of a defectless spherical cap of radius $r$ becomes

$$\Delta g_{cap}(\gamma) = -\frac{\Delta\tilde{\mu}}{4}x^2 + \frac{\lambda}{2}x + \frac{1}{1536}\frac{x^6}{r^4} + \frac{1}{4\gamma}x^2\left(\frac{1}{r}-1\right)^2.$$ (A28)

In the general case, the optimal radius of the shell is obtained from the condition $\partial\Delta g_{cap}/\partial r = 0$, yielding

$$r^2(r-1) - \frac{\gamma}{192}x^4 = 0.$$ (A29)

This equation shows that deviations from the preferred radius (i.e $r = 1$) are only expected for large domain sizes or large FvK numbers. The corresponding solution with positive second derivative can be obtained analytically, although its expression is a bit cumbersome. For small shells or FvK numbers, it can be well approximated by its two leading terms, yielding

$$r \simeq 1 + \frac{\gamma}{192}x^4.$$ (A30)

In the limit of large shells or Fvk numbers, the radius goes as

$$r \simeq \frac{1}{3} + \frac{1}{4}\left(\frac{\gamma}{3}x^4\right)^{1/3}.$$ (A31)

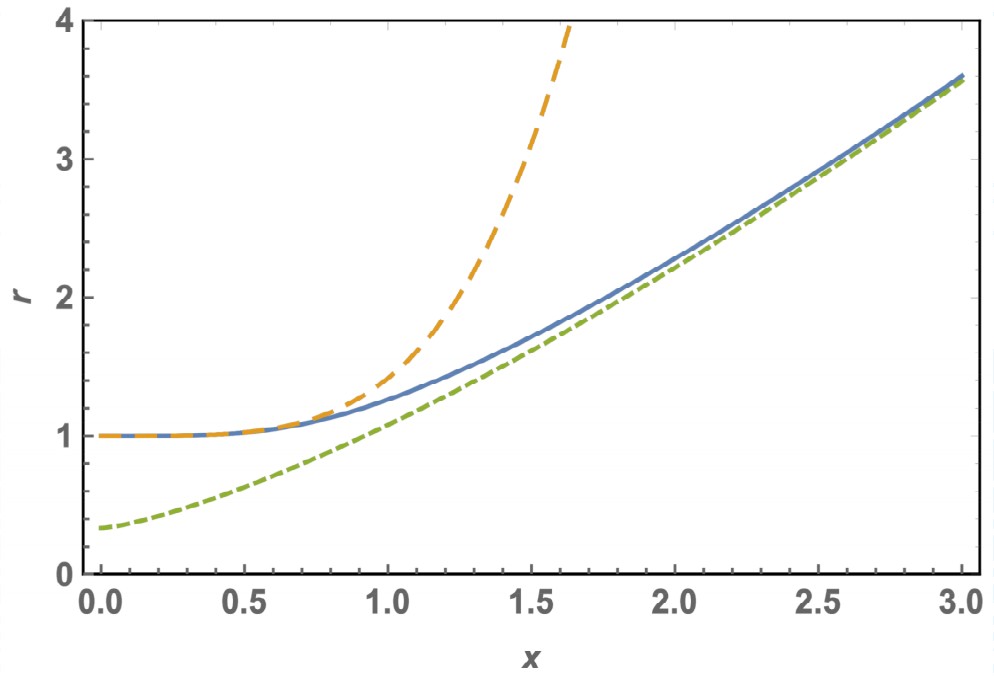

**Appendix 1—figure 2.** Scaled optimal radius $r$ of a spherical cap without defects as a function of the patch size $x$ for a FvK number $\gamma = 80$. The solid line is the exact solution, **Equation A29**, the short dashed line is the approximation for large shells or FvK, **Equation A31**, and the dashed line is the approximation for small shells or FvK, **Equation A30**.

In **Appendix 1—figure 2**, the curve $r$ vs $x$ is sketched as obtained from the complete solution of **Equation A29** (solid line). As can be seen, small caps adopt a curvature close to the one of the bending dominated case, $r \simeq 1$. Larger caps tend to flatten out. The dashed line shows the small shell approximation, **Equation A30**, while the short dashed line shows the assymptotic approximation given by **Equation A31**.

As the domain size increases, it becomes more favorable to release the elastic stress by the introduction of one or many 5-fold disclinations. The free enery of formation of a spherical shell with one central defect is

$$\Delta g_{d_1}(\gamma) = \Delta g_{cap}(\gamma) + \frac{x^2}{1152}\left(1 - \frac{3x^2}{2r^2}\right). \tag{A32}$$

which becomes favorable over the defectless case when $x/r \geq \sqrt{2/3}$. The formation energy of a spherical shell with $n$-defects is

$$\Delta g_{d_n}(\gamma) = \Delta g_{cap}(\gamma) + g_{s_1}(r) + g_{s_2} \tag{A33}$$

where

$$g_{s_1}(r) = \frac{x^2}{1152}\left(1 - \frac{3x^2}{2r^2}\right)\sum_{\alpha=1}^{n}\left(1 - \frac{x_\alpha^2}{x^2}\right)^2, \tag{A34}$$

and $g_{s_2}$ is given by **Equation A15**. The free energy of a ribbon of scaled width $w$ and length $l$ in the general case becomes

$$\Delta g_{rib}(\gamma) = -\frac{\Delta\tilde{\mu}}{4}x^2 + \frac{\lambda x^2}{2 w} + \frac{\lambda}{2\pi}w + \frac{9}{20480}x^2\left(\frac{w}{r}\right)^4 + \frac{1}{4\gamma}x^2\left(\frac{1}{r}-1\right)^2. \tag{A35}$$

The optimal width and radius of curvature of the ribbon, $w$ and $r$, are obtained by minimization that is, from $\partial\Delta g_{rib}/\partial w = 0$ and $\partial\Delta g_{rib}/\partial r = 0$, yielding

$$-\lambda \frac{x^2}{w^2} + \frac{\lambda}{\pi} + \frac{9}{2560} x^2 \frac{w^3}{r^4} = 0 \tag{A36}$$

and

$$-\frac{9\gamma w^4}{2560} - r^2(1-r) = 0. \tag{A37}$$

Finally, the free energies of a belt and a cylinder are

$$\Delta g_{belt}(\gamma) = -\frac{\Delta\tilde{\mu}}{4} x^2 + \lambda r + \frac{9}{327680} \frac{x^{10}}{r^8} + \frac{1}{4\gamma} x^2 \left(\frac{1}{r} - 1\right)^2 \tag{A38}$$

and

$$\Delta g_{cyl}(\gamma) = -\frac{\Delta\tilde{\mu}}{4} x^2 + \lambda r + \frac{1}{8\gamma} x^2 \left(1 + \left(\frac{1}{r} - 1\right)^2\right), \tag{A39}$$

respectively.

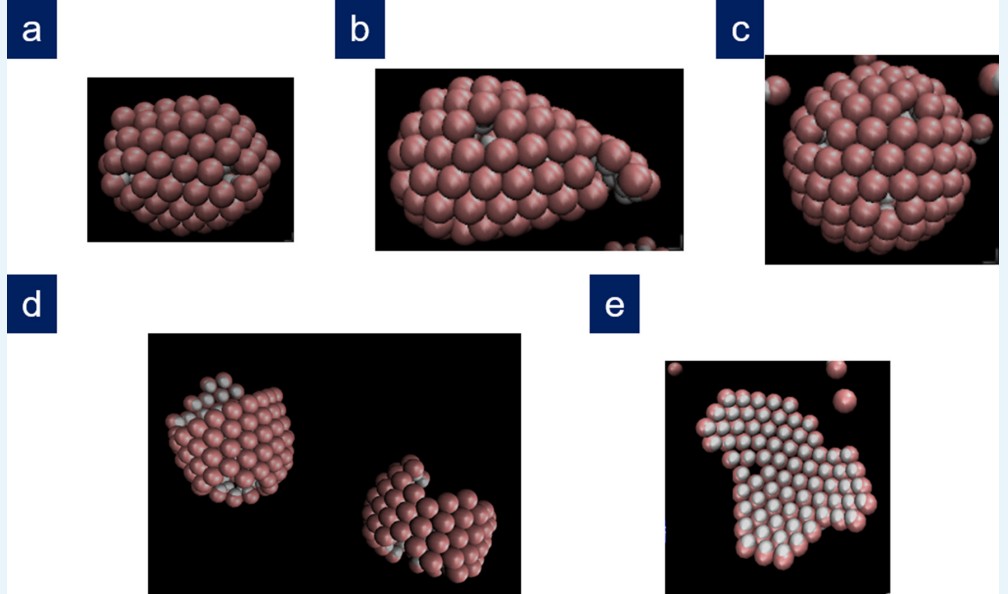

**Appendix 1—figure 3.** Other structures obtained in the simulations. (**a**) Bullet-shaped shell and (**b**) conical shell obtained in two different repetitions for $\gamma = 224$, $\lambda = 0.00069$, with $m = 36$, $n = 18$, $\nu = 1.345$, $\alpha = 0.4$ and $\rho = 0.02$; (**c**) T = 13 icosahedral shell obtained for $\gamma = 100$, $\lambda = 0.0005246$, with $m = 36$, $n = 18$, $\nu = 1.40$, $\alpha = 0.1$ and $\rho = 0.02$; (**d**) coexistence between a cylinder and a partial spherocylindrical shell obtained for $\gamma = 900$, $\lambda = 0.0005246$, with $m = 36$, $n = 18$, $\nu = 1.40$, $\alpha = 0.3$ and $\rho = 0.02$; (**e**) branched ribbon-like structure (***Köhler et al., 2016***) obtained at low values of the scaled line tension for $\gamma = 71$, $\lambda = 0.00031$, with $m = 36$, $n = 18$, $\nu = 1.47$, $\alpha = 0.1$ and $\rho = 0.0125$.

