## [Decision Letter]

**Acceptance summary:**

This study reports a theoretical analysis of the mechanical factors influencing the self-assembly (and mis-assembly) of protein subunits into viral capsids. This subject has been studied for a long time, but the present paper offers new insight regarding the importance of contribution from the protein line tension and preferred curvature. The main result of the study is (i) a universal phase diagram, which shows that viruses with high mechanical resistance cannot be self-assembled directly as spherical structures, and (ii) the ensuing suggestion of the need of a maturation step to stiffen the capsid after assembly. Inducing mis-assembly could be a promising route to hinder viral infections.

**Decision letter after peer review:**

Thank you for submitting your article "Triggering mis-assembly in viral capsid formation by elastic frustration" for consideration by *eLife*. Your article has been reviewed by three peer reviewers, and the evaluation has been overseen by a Reviewing Editor and Aleksandra Walczak as the Senior Editor. The reviewers have opted to remain anonymous.

The reviewers have discussed the reviews with one another and the Reviewing Editor has drafted this decision to help you prepare a revised submission.

Summary:

This paper combines a theoretical and a numerical study of the dependence of the self-assembly or mis-assembly process of viral capsids on parameters such as the chemical potential difference of protein subunits in and out of the capsid, the line tension, and the stretching and bending energies of the planar assembly, characterized by the Foppl-von Karman number. The main result of the study is (i) an universal phase diagram, which shows that viruses with high mechanical resistance cannot be self-assembled directly as spherical structures, and (ii) which justifies the need of a maturation step to stiffen the capsid after assemble. The authors suggest that this might be a promising route to hinder viral infections by inducing mis-assembly.

The reviewers found your study of the self-assembly (and mis-assembly) of protein subunits into capsids to be interesting and potentially important. Although the subject of viral capsid assembly has been studied for a long time, investigating the effect of the line tension, chemical potential and the preferred curvature of the protein assembly is a welcome addition to the existing literature.

The reviewers nevertheless have quite a few comments and questions, some of which are major points that should be addressed before a final decision can be made on the suitability of this submission of publication at *eLife*.

Essential revisions:

The reviewers nevertheless have some questions, some of which are major points that should be addressed before a final decision can be made on the suitability of this submission of publication at *eLife*.

1) The title of the paper focusses on mis-assembly, while most of the text discusses focus on assembly rather than mis-assembly. You should consider modifying the title to account for this.

2) Several important references are not cited. It is important to compare and contrast the results of the current study in some detail with earlier investigations. These include:

Defect in curved crystal:

– Azadi and Grason Phys. Rev. E 2016

Mechanisms of virus assembly:

– M.F. Hagan and D. Chandler, Biophys. J. 91, 42 (2006);

– O.M. Elrad, M.F. Hagan, Nano letters 8, 3850 (2008);

– H. Nguyen et al., J. Am. Chem. Soc. 131, 2606 (2009);

– I.G. Johnston et al., J. Phys. Condens. Matter 22, 104101 (2010);

– J.D. Perlmutter, M.F. Hagan, Ann. Rev. Phys. Chem. 66, 217 (2015).

Formation of clathrin-coated pits:

– R.J. Mashl et al., Biophys. J. 74, 2862 (1998);

– T. Kohyama et al., Phys. Rev. E 68, 061905 (2003);

– M. Giani et al., Biophys. J. 111, 222 (2016).

3) A table of numerical values is provided (Table 1). There are some questions about how the Young's modulus E is evaluated. In some references provided, in some of the references, E is not mentioned directly so the authors might have estimated it indirectly. Could the procedure for this evaluation be explained?

4) In order to determine the lowest energy sate, shape with similar number of subunits should be compared. Here, a rescaled size x is used, the definition of which appear to depend upon the shape considered. Hence a given x does not correspond to the same number of subunit for different shape. This should be corrected, and energies of shape with the same number of subunit should be compared.

5) The analytical results are based on a thermodynamic approach where the chemical potential is fixed, which implies a reservoir of particle. It is not clear how this compares with the numerical simulations. Is the chemical potential fixed during the assembly process in the simulations? This point should be made clear, and if the two approaches are based on different thermodynamic assumptions, how meaningfully can they be compared?

6) How does one relate the number of capsomers on the spherical surface, which is controlled by the chemical potential difference, and the number of particles. In the simulation, do we access the number of particles or the number of capsomers?

7) Regarding the bending-dominated regime, it seems a bit strange to rescale the chemical potential and line tension with the Youngs modulus, since this regime should contain the case where the Foppl-von Karman number (and the two-dimensional Young modulus Y) vanish, which implies that tilde(Δµ) and λ diverge. What is the optimal structure in this regime, where one does not expect differences been defect-free caps and caps with defects? This regime is not apparent in Figure 3A.

8) For the spherical cap with defect, how is the spatial distribution of the defect should be discussed? Does the energy used to compare with other shapes correspond to the lowest energy when defect localisation is optimised? This should be discussed in some details, since defect distribution is clearly an important factor influencing the energy of the cap.

9) The authors state that spherical capsids cannot self-assemble at large FvK. This statement is probably correct for strictly spherical capsids. However, aggregates could form as planar patches, then bend and form disclinations at sufficient size and line tension – see references on formation of clathrin-coated pits. Such structures are not accounted for in the analytical theory. Do they show up in the simulations, and if not, why not? Here again, the distribution of defect is a crucial factor that should be discussed.

10) Figure 3: Similar phase diagrams have been calculated by Schneider and Gompper. In their calculation, also structures consisting of several caps or several belts have been predicted. Are these structures relevant here as well? This should be discussed.

11) In the high concentration regimes "… many fragments of spherical capsids that cannot grow any further.…". Why can't they merge, rearrange their internal structure, and thereby reduce line-tension energy?

12) In the general case of non zero FvK numbers, the spontaneous radius of curvature plays an important role in the model, and this is discussed thoroughly by the authors. However, in the phase diagram, it seems that its discussion has been dismissed, or maybe its value has been fixed? In any case, some clarification on how the phase diagram depend on the value of spontaneous curvature is required. For example, it has been shown the reference by Castelnovo in 2017 that for small enough spontaneous curvature, cylinders should dominate. It is not clear how it compares to the many cases discussed here.

13) It is not clear why cylinders should dominate for large FvK numbers and large rescaled spontaneous curvature. Could the author elaborate on this point?

14) The authors should make more effort to recast their results in terms of experimentally useful quantities, so that it becomes apparent how to experimentally explore the phase diagram.

It is proposed that chemical or physical factors that increase the FvK number or reduce the line tension or the effective concentration are potential targets to prevent viral replication. It would be quite insightful to provide a more quantitative and practical version of this statement. Regarding the chemical potential, this statement could be turned into a capsid protein concentration at which viral assembly is expected to fail, given particular values of the other parameters. How does this concentration relates to expected concentration in cells? Regarding the mechanical parameter, in which range can they be expected to vary under the action of which factor, and would this be enough in practice to prevent capsid formation? You should provide more precise statement regarding how mis-assembly could be induced to hinder viral infections.

---

## [Author Response]

Essential revisions:The reviewers nevertheless have some questions, some of which are major points that should be addressed before a final decision can be made on the suitability of this submission of publication at eLife.1) The title of the paper focusses on mis-assembly, while most of the text discusses focus on assembly rather than mis-assembly. You should consider modifying the title to account for this.

We have changed the title of the work now entitled: “Shape selection and mis-assembly in viral capsid formation by elastic frustration”.

2) Several important references are not cited. It is important to compare and contrast the results of the current study in some detail with earlier investigations. These include:Defect in curved crystal:– Azadi and Grason Phys. Rev. E 2016Mechanisms of virus assembly:– M.F. Hagan and D. Chandler, Biophys. J. 91, 42 (2006);– O.M. Elrad, M.F. Hagan, Nano letters 8, 3850 (2008);– H. Nguyen et al., J. Am. Chem. Soc. 131, 2606 (2009)R;– I.G. Johnston et al., J. Phys. Condens. Matter 22, 104101 (2010);– J.D. Perlmutter, M.F. Hagan, Ann. Rev. Phys. Chem. 66, 217 (2015).Formation of clathrin-coated pits:– R.J. Mashl et al., Biophys. J. 74, 2862 (1998);– T. Kohyama et al., Phys. Rev. E 68, 061905 (2003);– M. Giani et al., Biophys. J. 111, 222 (2016).

We thank the reviewers for pointing out those relevant references. We have included all of them in the revised manuscript. More specifically, we have added a sentence at the beginning of the third paragraph to reference previous works on the mechanisms of virus assembly, including an additional reference (Hagan and Zandi, 2016). We have added another sentence to refer to the formation of clathrin-coated pits as well as carboxysomes and colloidosomes. Finally, we have included the reference to the work by Azadi and Hagan in the Discussion of crystal growth on spherical templates on the fourth paragraph.

3) A table of numerical values is provided (Table 1). There are some questions about how the Young's modulus E is evaluated. In some references provided, in some of the references, E is not mentioned directly so the authors might have estimated it indirectly. Could the procedure for this evaluation be explained?

The values of Young’s modulus reported on Table 1 are all, except that of SV40, provided by the references included in the caption. More specifically, the Young’s modulus of CCMV and λ capsid are listed in Table 1 of Mateu, 2012 and were evaluated using Finite Elements simulations by Michel et al., 2006 and Ivanoska et al., 2007, respectively. The Young’s modulus of λ procapsid was evaluated and reported by Sae-Ueng et al., 2014 from the spring constant measured by AFM using the standard thin shell formula. Finally, the Young’s modulus of SV40 was evaluated from the spring constant measured by AFM by van Rosmalen et al., 2018, using the standard thin shell formula k = 2.25 E h^2/R. We have included a sentence in the caption of Table 1 to clarify that. We have also added a reference to Ivanoska et al., 2004 and 2007 in the caption of Table 1 for the sake of completeness.

4) In order to determine the lowest energy sate, shape with similar number of subunits should be compared. Here, a rescaled size x is used, the definition of which appear to depend upon the shape considered. Hence a given x does not correspond to the same number of subunit for different shape. This should be corrected, and energies of shape with the same number of subunit should be compared.

Although, for simplicity and consistency, we write the equations and represent the energies for different shapes as a function of x, we do in fact determine the lowest energy state by comparing the energies for different shapes having the same area π *x*^2^, which is equivalent to having the same number of subunits. The variable x=ρ_0_/R_0_ is *always* defined as the radius of a circular patch of subunits, ρ_0_, divided by the spontaneous radius R_0_, so that its normalized area is π ρ_0_^2^/(4π R_0_^2^) = *x*^2^/4. For other shapes, in order to compare structures with the same number of subunits, what we do is to impose that they have the same area. Indeed, we use this requirement of equal areas to rewrite the scaled free energies of all shapes in terms of x. For example, for a ribbon of length L and width W, the normalized area is A_rib_/(4π R_0_^2^)=LW/(4π R_0_^2^)=lw/(4π). Thus, a ribbon with the same area (and thus, with the same number of subunits) as a circular patch implies lw=π *x*^2^, as written before Equation 5.

To clarify this point, we have added a sentence in the manuscript before Equation 2 and we have included this Discussion in the first paragraph of the bending dominated section and before Equation 19 in the Appendix.

5) The analytical results are based on a thermodynamic approach where the chemical potential is fixed, which implies a reservoir of particle. It is not clear how this compares with the numerical simulations. Is the chemical potential fixed during the assembly process in the simulations? This point should be made clear, and if the two approaches are based on different thermodynamic assumptions, how meaningfully can they be compared?

This is a very important point. While theory assumes a controlled chemical potential (equivalent to having a reservoir of particles), the simulations are done at fixed number of particles. This implies that in the simulation as the assembly proceeds, the concentration of the remaining free particles, and consequently the chemical potential, decrease. This may cause discrepancies when comparing both approaches, specifically if a precise quantitative comparison is intended. For this reason, we have not intended to reproduce with precision the values of the chemical potential or the borders of the phase diagram using the simulations. Nonetheless, for simulations with a large number of particles, the change in concentration and chemical potential due to the formation of a shell would be relatively small and can be neglected. But even if the chemical potential is not strictly constant in our simulations, the semi-quantitative comparison we were interested in is not altered since in any case we can compare structures with different relative concentrations and we can still say which structure will appear at the larger concentration, for example.

This important discussion and clarification about finite number of particles’ effects is now included at the end of section Simulation.

6) How does one relate the number of capsomers on the spherical surface, which is controlled by the chemical potential difference, and the number of particles. In the simulation, do we access the number of particles or the number of capsomers?

In the simulations we represent each subunit, i.e. (hexameric) capsomer, by a coarse-grained spherical particle. The simulations are performed with a fixed total number of subunits N. The initial configuration consists of a small spherical cap of 19 subunits with the spontaneous radius R_0_ as initial seed, and the remaining N-19 subunits are randomly distributed inside a fixed-sized simulation box. Thus, the total concentration of subunits (or capsomers) is fixed in a given simulation. Please note that in the simulations we are only inserting an initial seed, but we are not forcing the capsomers to be in any kind of spherical or cylindrical template. Instead, the subunits spontaneously self-assemble into the energetically-preferred shape and size. Thus, it is the total number of subunits (i.e. capsomers) that is controlled, but it is the time evolution of the seeded system, without any bias or template, which determine how many subunits end up forming the energetically preferred structure.

7) Regarding the bending-dominated regime, it seems a bit strange to rescale the chemical potential and line tension with the Youngs modulus, since this regime should contain the case where the Foppl-von Karman number (and the two-dimensional Young modulus Y) vanish, which implies that tilde(Δµ) and λ diverge. What is the optimal structure in this regime, where one does not expect differences been defect-free caps and caps with defects. This regime is not apparent in Figure 3A.

The situation of a zero Young’s modulus is unrealistic, since it would correspond to a mechanical unstable material that could be stretched or compressed without any energy cost. Thus, we are always considering a finite Y to normalize the chemical potential and the line tension. Rather than with a negligible Young modulus, the bending-dominated regime, corresponding to the limit in which the Foppl-von Karman number vanish, is achieved by having a very large bending modulus compared to the Young modulus times R_0_^2^. In this case the structure cannot be bend out from its preferred curvature R_0_ since it would cost a lot of bending energy. This is the case depicted in Figure 3A. On the other hand, the limit in which Y vanishes would imply a structure that does not pay any stretching energy. In this case, there is almost no difference for the structure to incorporate defects or not, but also only the first two terms of Equation 1 would remain. Thus, at equal areas, partially formed caps with the preferred curvature would prevail over the ribbon due to the smaller edge energy. In addition, in the limit of vanishingly small Y, the minimum due to elastic frustration will only occur at very large patch areas, exceeding the values corresponding to a closed shell. Euler’s theorem states that at least 12 defects are required to have a closed shell. So, in the bending dominated regime and in the limit Y very small, corresponding to very large values of the scaled ∆µ and λ, the optimal structure would still be a closed spherical cap with defects. In fact, we have shown in the Appendix that the defectless spherical cap is always metastable. Therefore, all realistic situations are already accounted for in the bending-dominated scenario depicted in Figure 3A.

8) For the spherical cap with defect, how is the spatial distribution of the defect should be discussed? Does the energy used to compare with other shapes correspond to the lowest energy when defect localisation is optimised? This should be discussed in some details, since defect distribution is clearly an important factor influencing the energy of the cap.

In Li et al., 2018, the energy of an incomplete cap with one defect placed at an arbitrary location is calculated. It is found that the Gaussian curvature attracts the disclination to the center of the cap while the defect self-energy pushes it towards the boundary. The net result is that the minimum energy corresponds to the defect located off the center of the cap. In our calculation we have used the approximation in which the defect is located at the center. However, we have numerically verified that this approximation introduces only a very small error, of less than 0.25% , in our calculations for the scaled energy. This means that not noticeable effect is observed when the exact expression with the off-center defect is considered. On the other hand, for a cap with more than one disclination, the repulsive interaction among them, as well as their repulsion with the free boundary, contribute to distribute them regularly, as considered in our calculation. When the cap finally closes, the maximal number of disclinations is 12, their regular distribution locates them along the vertices of an icosahedron, which corresponds to the lowest energy configuration.

9) The authors state that spherical capsids cannot self-assemble at large FvK. This statement is probably correct for strictly spherical capsids. However, aggregates could form as planar patches, then bend and form disclinations at sufficient size and line tension – see references on formation of clathrin-coated pits. Such structures are not accounted for in the analytical theory. Do they show up in the simulations, and if not, why not? Here again, the distribution of defect is a crucial factor that should be discussed.

In our model, the condition for planar patches to assemble is ∆µ(tilde)>=1/γ, but only if the preferred curvature is zero, otherwise, the assembly of cylinders will be preferable due to a smaller bending energy. If the preferred curvature is zero, there would be no reason to bend out of the plane, which would also cost a lot of stretching energy, unless an additional external mechanism forcing the bending is present (which is the case in clathrin-coated pits). Thus, in our case, at large FvK the theory does not predict spherical capsids as preferred structure and we indeed have not found them in the simulations performed at large FvK.

10) Figure 3: Similar phase diagrams have been calculated by Schneider and Gompper. In their calculation, also structures consisting of several caps or several belts have been predicted. Are these structures relevant here as well? This should be discussed.

In Schneider and Gompper’s study, the crystalline domains appear on a spherical fluid vesicle acting as a template. Their phase diagrams are constructed in terms of the relative area covered by the crystalline phase, considering the possibility that it may consist of independent belts or caps growing on the *same* vesicle. In our study, on the other hand, there is no pre-existing vesicle or template and the assembled caps or belts form *independently* of each other depending on the chemical potential and not by imposing a fixed target area. What it is possible, however, is that during the assembly process, shapes similar to a central cap with ribbon-like protrusions (similar to a flower) or branched ribbon-like structures (Appendix 1—figure 3E) may appear. These more complex shapes have not been considered in our study.

11) In the high concentration regimes "… many fragments of spherical capsids that cannot grow any further.…". Why can't they merge, rearrange their internal structure, and thereby reduce line-tension energy?

The kinetic trapping mechanism we discuss in that part of the text, is due to the formation of many fragments, that in a simulation with a fixed number of subunits causes that the concentration of the remaining free monomers eventually reduces to a value that is smaller than the critical concentration needed to continue the assembly of the fragments (this is related to point 5 above). Of course, these fragments would be subjected to Brownian motion and may eventually meet at the right orientations and merge, reducing the line tension. But it is very unlikely that many fragments containing a different number of subunits and defects end up forming a perfect closed shell. (It would be similar to the formation of a nice single crystal by the coarsening of multiple small crystalline domains grown independently.) In any case, the kinetic trapping mechanism and the possibility of forming the right structure at longer times by coarsening, go beyond of the present work and will be discussed in a future publication, as mentioned in the manuscript.

12) In the general case of non zero FvK numbers, the spontaneous radius of curvature plays an important role in the model, and this is discussed thoroughly by the authors. However, in the phase diagram, it seems that its discussion has been dismissed, or maybe its value has been fixed? In any case, some clarification on how the phase diagram depend on the value of spontaneous curvature is required. For example, it has been shown the reference by Castelnovo in 2017 that for small enough spontaneous curvature, cylinders should dominate. It is not clear how it compares to the many cases discussed here.

The phase diagrams (Figure 3) have been drawn using scaled variables. The scaled line tension is λ=Λ/(R_0_Y), where R_0_ is the spontaneous radius of curvature. We can make contact with Castelnovo results noticing that small spontaneous curvature means a large R_0_ which for a given line tension Λ implies a small value of λ which is the region where belts tend to appear. On the other hand, the phase diagram show that cylinders appear at Δμ(tilde)>=1/(2γ), which, after using the definition of the scaled quantities can be rewritten as Δμ>=κ a_1_/(2R_0_^2^). In other words, cylinders appear more easily (smaller Δμ required) for larger values of R_0_, in complete agreement with Castelnovo results. This is now clarified and discussed in the manuscript.

13) It is not clear why cylinders should dominate for large FvK numbers and large rescaled spontaneous curvature. Could the author elaborate on this point?

At large FvK numbers, corresponding to the regime where stretching dominates over bending, cylinders have the advantage of not having any stretching energy cost (i.e. a flat sheet of hexamers can be bent into a cylinder without any stretching). A cylindrical structure having a radius equal to the spontaneous radius R_0_, i.e. r=1, will minimize the bending penalty and will have a free energy of formation, according to Equation 12, that decreases unboundedly with the size when Δμ(tilde)>=1/(2γ). In other words, once the formation of a cylinder becomes more favorable than free capsomers, it will continue growing without limit decreasing indefinitely its free energy of formation without paying any stretching cost, thus overcoming the energetic gain of any finite sized structure. This will be the case when Δμ(tilde)>=1/(2γ). The larger the γ (FvK), the smaller the Δμ(tilde) required for this to occur and therefore, regions where finite sized structures where preferred start to be devoured by the region where cylinders dominate (purple regions in Figure 3).

This discussion has been incorporated in the manuscript, just before the simulations section.

14) The authors should make more effort to recast their results in terms of experimentally useful quantities, so that it becomes apparent how to experimentally explore the phase diagram.It is proposed that chemical or physical factors that increase the FvK number or reduce the line tension or the effective concentration are potential targets to prevent viral replication. It would be quite insightful to provide a more quantitative and practical version of this statement. Regarding the chemical potential, this statement could be turned into a capsid protein concentration at which viral assembly is expected to fail, given particular values of the other parameters. How does this concentration relates to expected concentration in cells? Regarding the mechanical parameter, in which range can they be expected to vary under the action of which factor, and would this be enough in practice to prevent capsid formation? You should provide more precise statement regarding how mis-assembly could be induced to hinder viral infections.

The main goal of the work was to provide a general understanding of the assembly phase diagram that could guide future experiments. But in order to make precise quantitative predictions that could be experimentally tested, we need a few parameters that, as far as we know, have unfortunately not been measured for real viruses, yet. For instance, the critical concentration c* is needed in order to make a quantitative prediction of the capsid protein concentration at which viral assembly is expected to fall. Without accurate evaluation of these parameters it is too adventurous to make reliable quantitative predictions. We hope that our work will stimulate future experimental work aimed at measuring these important parameters for different viruses of interests. In any case, we have added a paragraph in the Discussion to make a closer connection between the physical parameters controlling the assembly and experiments.

The added paragraph reads:

“Experimentally, the chemical potential can be tuned by the total protein concentration or by the addition of crowding agents. The line tension (which depends on the strength of the binding interaction), could be modified by the temperature, the pH and the salt concentration. The bending rigidity and spontaneous radius of curvature are also presumably controlled by pH and the presence, concentration, and nature of ions or auxiliary proteins in solution. Further experimental and theoretical investigations are required to make a precise quantitative connection between the physical parameters controlling the assembly and experiments.”